# Narrow Transformer:
# Mono-lingual Code SLM for Desktop

## Abstract

This paper presents NT-Java-1.1B, an open-source specialized code language model built on StarCoderBase-1.1B[1], designed for coding tasks in Java programming. NT-Java-1.1B achieves state-of-the-art performance, surpassing its base model and majority of other models of similar size on MultiPL-E (Cassano et al., 2022) Java code benchmark. While there have been studies on extending large, generic pre-trained models to improve proficiency in specific programming languages like Python, similar investigations on small code models for other programming languages are lacking. Large code models require specialized hardware like GPUs for inference, highlighting the need for research into building small code models that can be deployed on developer desktops. This paper addresses this research gap by focusing on the development of a small Java code model, NT-Java-1.1B, and its quantized versions, which performs comparably to open models around 1.1B on MultiPL-E Java code benchmarks, making them ideal for desktop deployment. This paper establishes the foundation for specialized models across languages and sizes for a family of NT Models.

## 1 Introduction

The state-of-the-art code models, capable of understanding and generating code in numerous programming languages, are revolutionizing the way enterprises approach software development. With the ability to understand and generate code across a vast array of programming languages, these code models offer a significant boost in productivity. However, the one-size-fits-all approach of these generic multi-lingual code models often falls short in meeting the nuanced requirements of project-level coding tasks in an enterprise, which tend to be language-specific. This has led to the development of Narrow Transformers (NTs), specialized models further trained on a particular programming language, offering a more efficient solution for enterprises. These NTs are designed to optimize performance for a specific programming language, balancing the trade-offs between model size, inferencing cost, and operational throughput. As demand for tailored solutions grows, we can expect a surge in NT development, providing the precision and efficiency required by enterprise projects.

However, in practice, the substantial economic cost associated with training and fine-tuning large code models renders language model experiments prohibitively expensive for most researchers and organizations. Additionally, deploying these massive models in everyday scenarios, such as on personal computers, proves either inefficient or unfeasible. These challenges emphasize the importance of shifting focus to explore Narrow Transformer approach on powerful yet smaller code language models (code SLMs). Consequently, we developed a Narrow Transformer for Java within a smaller parameter range (i.e., 1.1B), suitable for desktop deployment and democratizing code model experiments.

## 2 Related Work

Codex-12B (Chen et al., 2021) was built by extending pre-training of GPT (which contains strong natural language representations), with 159 GB of unique Python files under 1MB, from public software repositories hosted on GitHub. Codex exhibits its highest proficiency in Python; however, it

---

[1]https://huggingface.co/bigcode/starcoderbase-1b

also demonstrates competence in over twelve additional programming languages. CodeGen-Mono-350M/2.7B/6.1B/16.1B (Nijkamp et al., 2023b) were built by further pretraining CodeGen-Multi-350M/2.7B/6.1B/16.1B (which were trained with multi-lingual datasets comprising code from C, C++, Go, Java, JavaScript, and Python) with the mono-lingual dataset BIGPYTHON that contains public, non-personal, permissively licensed Python code from GitHub. CodeGen-Mono outperformed CodeGen-Multi on Python as per the HumanEval benchmark. In addition, the next generation model in CodeGen family, such as, CodeGen25-7B-mono (Nijkamp et al., 2023a) outperformed CodeGen25-7B-multi only in python language but underperformed in rest of the programming languages in MultiPL-E benchmark. StarCoder-15.5B (Li et al., 2023) was built by extending pretraining of StarCoderBase-15.5B (which was trained with multi-lingual datasets comprising code from 80+ programming languages) with a Python subset of 35B tokens from the StarCoderBase training data. StarCoder outperformed StarCoderBase on Python as per the HumanEval benchmark. In the evaluation of StarCoder and StarCoderBase on 19 programming languages with MultiPL-E datasets, StarCoder outperformed StarCoderBase on Python, underperformed on 9 programming languages, and despite being further trained only on Python, it still outperformed StarCoderBase on 9 other programming languages. CodeLlama-PYTHON-7B/13B/34B/70B (Rozière et al., 2023) were built by extending pre-training of CodeLlama-7B/13B/34B/70B (which were trained on 500B tokens of code data, except CodeLlama-70B, which was trained on 1T tokens) on 100B tokens of python heavy dataset with a composition of Python, multi-lingual code, natural language related to code and natural language at the proportions of 75%, 10%, 10%, 5% respectively. CodeLlama-PYTHON outclasses CodeLlama on Python on MultiPL-E benchmarks, but it is not consistent on rest of the languages. While there are speculations explaining this inconsistency, it is generally understood that although extending pretraining of multi-lingual code foundation models with dataset from a specific programming language does not guarantee performance improvement in other programming languages, it still guarantees performance improvement in that programming language. Hence, building a model like StarCoder using a specific programming language dataset can improve proficiency in that programming language. Enterprise projects are adopting either these pre-trained generic multi-lingual code models or python-trained multi-lingual code models to augment their project coding tasks. AI-mature enterprises are adopting these models as foundation models to further train with their project code base for better augmentation. However, if there is a pre-trained code model further trained on enterprise project's required programming language, then the enterprise project can use that language-specific model and can further train with their project code base for better augmentation. Due to the widespread adoption of Java in enterprise-level projects, this paper illustrates the development of such a pre-trained code model specialized on Java.

Small Language Models (SLMs) will pivot the focus of AI community in enterprise and consumer solutions. These models stand out for their ability to be deployed on end-user devices, such as personal computers and smartphones, even without a GPU. This enables large-scale deployment while ensuring data privacy and security. Significant examples in the present scenario of code SLMs include SantaCoder-1.1B (Allal et al., 2023), Phi-1 (Gunasekar et al., 2023), DeciCoder-1B[2], StarCoderBase-1.1B, WizardCoder-1B-V1.0 (Luo et al., 2023), DeepSeek-Coder-1b-base (Guo et al., 2024) and Refact-1.6B[3]. All these state-of-the-art models around 1B size are multi-lingual code models, indicating that no considerable work has been done towards extending training of multi-lingual code SLMs in building language-specific code SLMs.

## 3 DATASETS

The foundation model identified for our experiment was StarCoderBase-1.1B. Enterprise projects shortlist the candidate code models for adoption of coding tasks based on their licenses, their training data, etc. Utilizing additional dataset, such as pretraining dataset from any model other than StarCoderBase, to extend the pretraining of StarCoderBase-1.1B would complicate the process of shortlisting the further trained StarCoderBase-1.1B model (NT-Java-1.1B) for any enterprise adoption, due to the concerns on licensing. Hence, a subset of StarCoderData[4], which is a curated dataset from The Stack v1[5] used for StarCoderBase training, was considered for building NT-Java-1.1B.

---

[2] https://huggingface.co/Deci/DeciCoder-1b

[3] https://huggingface.co/smallcloudai/Refact-1_6B-fim

[4] https://huggingface.co/datasets/bigcode/starcoderdata

[5] https://huggingface.co/datasets/bigcode/the-stack

The rationale behind building Python-trained models such as Codex, CodeGen-Mono, StarCoder, and CodeLlama-PYTHON might be the popularity of Python and the availability of the greater volume of Python code in the pretraining dataset compared to other programming languages. While the Python dataset in the StarCoderBase training dataset is 35B Python tokens, the Java dataset is around 22B tokens, which is still a considerable size. This Java dataset from StarCoderData was used for training NT-Java-1.1B.

## 4 MODEL TRAINING

### 4.1 DATA PREPROCESSING

For data preprocessing, we employed the Megatron-LM framework. The NT-Java-1.1B uses the StarCoderBase tokenizer of type GPT2BPETokenizer (byte-level Byte-Pair-Encoding) and its vocabulary of 49,152 tokens. No additional tokens were added to this vocabulary. The Java dataset comprises 87 parquet files, which were converted into a single file and passed through the Megatron pre-processing module to get the corresponding .bin and .idx files. These files were used for model training. The pre-processing module also performs tokenization and adds an <EOD> token at the end of each Java sample.

### 4.2 MODEL ARCHITECTURE

NT-Java-1.1B, similar to StarCoderBase-1.1B, is a decoder-only Transformer model with Multi-Query Attention (Shazeer, 2019), which uses FlashAttention. This speeds up the attention computation and reduces the training time of the model. The hyper-parameters for the architecture can be found in Table 1.

Table 1: Model architecture of NT-Java-1.1B.

| Hyperparameter | NT-Java |
|---|---|
| Hidden size | 2048 |
| Intermediate size | 8192 |
| Max. position embeddings | 8192 |
| Num. of attention heads | 16 |
| Num. of hidden layers | 24 |
| Attention | Multi-query |
| Num. of parameters | $\approx$ 1.1B |

### 4.3 TRAINING DETAILS

NT-Java-1.1B was trained using the Megatron-LM Framework. The training began with StarCoderBase-1.1B, serving as the initial checkpoint, to build its Java variant. In our experiments, we utilized a context length of 8192 tokens for tasks involving the Next token prediction and the Fill-in-the-Middle (FIM) (Bavarian et al., 2022) objective. The PyTorch Distributed framework was employed, with data parallelism strategy. We chose bf16 precision and the Adam optimizer (Kingma & Ba, 2015) with $\beta1 = 0.9$, $\beta2 = 0.95$, and $\epsilon = 10^{-8}$, along with a weight decay of 0.1.

EXPERIMENTAL SETTINGS

In this study, we delve into the impact of extending pretraining of StarCoderBase-1.1B for Java using two key objectives: Next token prediction and Fill-in-the-Middle.

Experiment 1 - Next token prediction objective: We conducted training over 100,000 steps (equivalent to 5 epochs) with a batch size of 1 million tokens. The learning rate commenced at $4 \times 10^{-4}$ and underwent cosine decay, reaching a minimum of $4 \times 10^{-6}$ with 1,000 iterations of linear warmup. A global batch size of 180 facilitated the training process, which spanned 12 days. Model checkpoints were saved every 1,000 steps for subsequent evaluation.

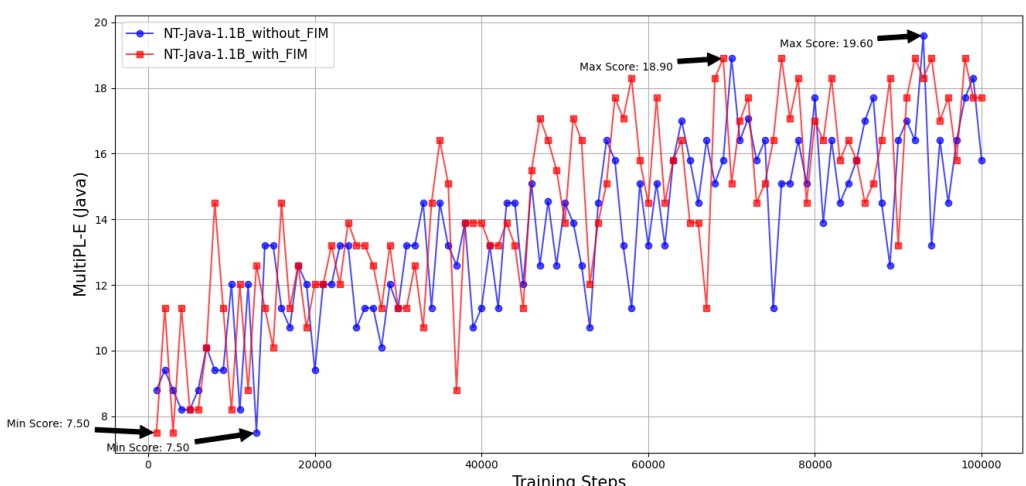

Figure 1: MultiPL-E Scores of NT-Java-1.1B trained with and without FIM.

Experiment 2 - Fill-in-the-Middle: We repeated Experiment 1 along with FIM training objective. The FIM rate was set to 50%. The FIM dataset was evenly split into two components, SPM (Suffix-Prefix-Middle) and PSM (Prefix-Suffix-Middle).

Observation from Experiment 1 & 2: Without FIM training objective, the model's infilling capability diminished significantly, with FIM scores approaching nearly zero (Table 2), despite the base model's inherent infilling capability. While training with FIM objective, we observed a minor decrease in MultiPL-E metrics (approximately 0.7%) compared to the model trained without FIM objective, but the model retained its proficiency in infilling tasks. The comparative performance of the models throughout the training are illustrated in Figure 4.3.

Table 2: Experimental results with and without FIM.

| Model | FIM | HumanEval-FIM (Java) | MultiPL-E (Java) |
|---|---|---|---|
| NT-Java-1.1B (Experiment 1) | No | 0.01 | 19.6 |
| NT-Java-1.1B (Experiment 2) | Yes | 0.67 | 18.9 |

Experiment 2.1 - Fill-in-the-Middle: We extended training from Experiment 2 for 20,000 steps (1 epoch) more as the evaluation scores were in an upward trend. The learning rate commenced at $4\times10^{-6}$ and underwent cosine decay, reaching a minimum of $4\times10^{-7}$ with 1,000 iterations of linear warmup. We did not intend to continue further training as the model converged with no significant decrease in loss.

## 4.4 POST TRAINING

The NT-Java-1.1B model has bf16 precision and occupies a total size of 2.27 GB. After the development of the NT-Java-1.1B model, efforts were directed towards the development of quantized models that are tailored to operate on developer desktops. These models were designed to be more compact in size without substantially sacrificing their accuracy, and to be compatible with CPU-based inference frameworks. To achieve this, we built quantized variants of the NT-Java-1.1B model in GGUF[6] format for frameworks like Ollama[7], GPT4ALL[8] and LM Studio[9]. The quantized versions

---

[6]https://github.com/ggerganov/ggml/blob/master/docs/gguf.md
[7]https://github.com/ollama/ollama
[8]https://github.com/nomic-ai/gpt4all
[9]https://github.com/lmstudio-ai

of the models (NT-Java-1.1B-GGUF) are available in a range from 2-bit to 8-bit, with their overall sizes spanning from 511 MB to 1.32 GB correspondingly.

## 4.5 COMPUTE

NT-Java-1.1B was trained with 6 A100 80 GB GPUs on a single-node GPU cluster. The training process remained stable overall, with only a few restarts.

## 5 EVALUATION

This section presents evaluation of our proposed coding SLM to assess its capabilities in code generation and infilling tasks.

### 5.1 MULTIPL-E

In our initial assessment, we evaluated the performance of the model from Experiment 2.1 on Java code generation tasks by utilizing the widely recognized benchmark, MultiPL-E. We calculated the pass@1 metric for this benchmark utilizing the BigCode Eval Harness[10], ensuring the hyperparameter values were aligned with the established norms of the Big Code Models Leaderboard[11]. NT-Java-1.1B demonstrated a pass@1 score that surpassed its base model and its 3B variant, as detailed in Table 3. Furthermore, our model's performance surpassed majority of the base models within a similar parameter range, such as Phi-1, SantaCoder-1.1B, DeciCoder-1B, OctoGeeX-7B, StableCode-3B-alpha, WizardCoder-1B-V1.0 and CodeGen25-7B-mono, on the Big Code Models Leaderboard.

Table 3: Pass@1 results on MultiPL-E.

| Model | Java |
| --- | --- |
| StarCoderBase-1.1B | 14.2 |
| StarCoderBase-3B | 19.25 |
| **NT-Java-1.1B** | **20.2** |

### 5.2 FILL-IN-THE-MIDDLE BENCHMARK

Subsequently, we conducted an evaluation of the model's capabilities on the single-line code infilling task, utilizing the benchmark established in the SantaCoder. This benchmark gauges the model's proficiency in completing a single line of Java code within HumanEval solutions, using the 'line exact match' accuracy as the evaluation metric. Our analysis indicates that our model delivers results that are on par with the foundational model, StarCoderBase-1.1B, showcasing comparable performance, as outlined in Table 4.

Table 4: HumanEval-FIM scores.

| Model | Java |
| --- | --- |
| StarCoderBase-1.1B | 0.71 |
| **NT-Java-1.1B** | **0.67** |

### 5.3 COMPUTATIONAL CAPABILITIES

Furthermore, we evaluated the model's performance in terms of its efficiency and resource utilization. Our analysis (Table 5) indicates that our NT-Java quantized models achieve an optimal balance

---

[10]https://github.com/bigcode-project/bigcode-evaluation-harness
[11]https://huggingface.co/spaces/bigcode/bigcode-models-leaderboard

between accuracy and resource utilization, making them a suitable candidate for deployment in resource-constrained environments. For the computation of the MultiPL-E scores of the quantized variants, we employed the 'load in 4-bit' and 'load in 8-bit' parameters within the BigCode Eval Harness.

Table 5: Accuracy and resource utilization.

| Model | Pass@1 (Java) | Size (GB) |
|---|---|---|
| StarCoderBase-1.1B | 14.2 | $\approx 2.27$ |
| **NT-Java-1.1B_Q4** | **15.1** | **0.76** |
| **NT-Java-1.1B_Q8** | **17.7** | **1.23** |
| StarCoderBase-3B | 19.25 | $\approx 6.1$ |
| **NT-Java-1.1B** | **20.2** | **2.27** |

As a last step, we conducted qualitative evaluations through user studies. Professional developers and coding enthusiasts were invited to interact with our model, providing insights into the model's usability, the relevance of its code suggestions, and its adaptability to user prompts. The feedback collected underscores the model's practical utility and its potential to streamline coding workflows.

## 6 CONCLUSION

In this technical report, we outlined the rationale and training approach used to develop NT-Java-1.1B, a small language model trained specifically on Java code. We evaluated NT-Java-1.1B across various coding tasks and compared its performance against models with similar parameters. Our findings indicate that NT-Java-1.1B is competitive with or outperforms other Code SLMs in this parameter range in Java programming tasks.

This study demonstrates the successful achievement of its objective of enhancing the efficiency of a code SLM for a particular programming language by training it further with a subset of its dataset for that language. While the research employed the StarCoderBase-1.1B model and its Java language dataset, other SLMs and their associated programming language datasets can yield comparable experimental outcomes.

The release of NT-Java-1.1B and its variants aims to democratize code foundation models, making them accessible for deployment in memory-constrained environments such as developer desktops and laptops. By adhering to the principles of the OpenRAIL-M[12] and by open-sourcing the corresponding scripts on GitHub, we hope to enable both the research and developer communities to experiment and adopt code SLMs.

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
