# OpenReview forum: "Narrow Transformer: Mono-lingual Code SLM for Desktop"
_ICLR.cc/2025/Conference — Submitted to ICLR 2025_

### Official Review · Reviewer_7phg · 2024-10-22

**Soundness:** 3
**Presentation:** 3
**Contribution:** 1
**Rating:** 3
**Confidence:** 5

**Summary:**

The authors propose to train a small code model called "NT-Java-1.1B" specialized for Java. Their small model achieves better performance on Java than the StarCoderBase-1.1B counterpart which they use as the initial starting checkpoint. The small model size allows the model to efficiently run on laptop devices or other edge hardware thereby increasing productivity of developers writing Java code.

**Strengths:**

1. The paper is easy to read and well-written and easy to follow.
2. The authors release their models under the OpenRAIL-M license which allows both research and commercial use.

**Weaknesses:**

1. There is no novelty in the paper.
2. Evaluations are missing for the GGUF model.

**Questions:**

The paper is easy to read and well-written although there are some gaps:
1. There is no reason to talk about the binary .bin and .idx files format in Megatron in section 4.1
2. line 130 says that the model uses FlashAttention, however, it should be noted that FlashAttention is not a part of the model, its just a training time optimization.
3. The paper mentions using `load_in_4bit` and `load_in_8bit` arguments from HuggingFace APIs are used for evaluation however, as far as I am aware, `load_in_8bit` uses [LLM.int8()](https://arxiv.org/abs/2208.07339) algorithm to quantize the model. It would be better to use a better algorithm like GPTQ or AWQ to report the accuracy. The `load_in_4bit` argument uses FP4/NF4 quantization from the bitsandbytes library. While these are easily usable from a user's perspective, it would be nice to have evaluation numbers with the GGUF model.
4. Table 3 and 4 can be combined into a single table and there is no need for 2 different tables.
5. There is no comparison with Llama-3.2 1B model. It would be better to add that comparison as a baseline.
6. It might be better to train a MoE model with less activated parameters for efficiency I think. Something like 3B full parameters with ~500-800M activated parameters for better model accuracy while still being efficient. However, I understand that this might be unfeasible due to compute restrictions.

---

### Official Review · Reviewer_NJgP · 2024-10-27

**Soundness:** 2
**Presentation:** 2
**Contribution:** 1
**Rating:** 3
**Confidence:** 5

**Summary:**

This paper introduces NT-Java-1.1B, a specialized code language model designed for Java programming, built on StarCoderBase-1.1B. It aims to propose small, efficient code models that can be deployed on developer desktops. NT-Java-1.1B achieves state-of-the-art performance on the MultiPL-E **Java** code benchmark, surpassing its base model and other similar-sized models.

**Strengths:**

1. Investigating small and efficient code models is meaningful.

**Weaknesses:**

1. This paper merely fine-tunes an open-source base model (StarCoderBase-1.1B) using part of public data (The Stack v1) and existing training methods. The result is evaluated on only part of the MultiPL-E benchmark. This paper offers no new technical contributions, lacks thorough experimental analysis, and provides no insights.
2. The term "Narrow Transformers" is neither defined nor has any related references. It seems like a concept created by the authors. I'm confused about how this concept differs from small LLMs or efficient LLMs. Could the authors explain this?
3. The experimental results are poor. The fine-tuning of the base model StarCoderBase-1.1B for JAVA yields results that are basically similar to StarCoderBase-1.1B itself, while StarCoderBase-1.1B is capable of handling multilingual tasks instead of just JAVA.
4. This paper only contains 6 pages.

**Questions:**

No questions.

---

### Official Review · Reviewer_2sSg · 2024-10-29

**Soundness:** 2
**Presentation:** 2
**Contribution:** 1
**Rating:** 3
**Confidence:** 4

**Summary:**

This work introduced the NT-Java-1.1B model, detailing its development process and evaluation results. NT-Java-1.1B is developed based on the StarCoderBase-1.1B model and trained on a subset of StarCoderData. Evaluation results indicate that NT-Java-1.1B outperforms StarCoderBase-3B in pass@1 performance on the MULTIPL-E benchmark, while it scores lower than StarCoderBase-1.1B on HumanEval-FIM (Java).

**Strengths:**

This work provides a detailed introduction to the training process of NT-Java-1.1B and examines the impact of FIM (Fill-in-the-Middle) training, which can offer reference value for developing other SLMs.

**Weaknesses:**

+ This work’s novelty is limited, as it builds on the StarCoderBase-1.1B model [1] using training data from StarCoderData [2] and applies established methods such as Next Token Prediction [3] and Fill-in-the-Middle [4]. While it provides an application of these methods, it does not introduce new improvements or substantial contributions.

+ The experiments in this work are insufficient, as there are too few baseline models compared to NT-Java-1.1B; only StarCoderBase-1.1B and StarCoderBase-3B are included, lacking comparisons with other models.

[1] StarCoder: may the source be with you!

[2] https://huggingface.co/datasets/bigcode/starcoderdata

[3] Improving Language Understanding by Generative Pre-Training

[4] Efficient training of language models to fill in the middle

Minor issues: In Line 154, a subsection number is missing.

**Questions:**

At the end of the Evaluation section, the authors mentioned that they conducted qualitative evaluations through user studies, but details and results of this evaluation are not provided. Could you share more comprehensive results for this part of the evaluation?

The authors have trained a model and indicated that its performance has improved. Could you elaborate further on the novelty in your work and clarify if there are any unique contributions?

---

### Official Review · Reviewer_YFW9 · 2024-11-05

**Soundness:** 3
**Presentation:** 2
**Contribution:** 1
**Rating:** 1
**Confidence:** 5

**Summary:**

The authors introduce NT-Java, a narrowly fine-tuned code language model of StarCoder's bade model of 1B parameters especially suited for edge applications on smaller devices. The model is finetuned on the Java-portion of the Stack, and training is performed with Nvidia's Megatron LM framework. The model's performance is evaluated on a next token prediction objective, and a fill-in-the-middle objective. Quantized versions of the tuned model are made available for wider use.

**Strengths:**

The paper is written with a coherent story-line, which leads one through the entire paper, and does a very good job in describing its experiments as well as the exact setup used. The reviewer is highly confident that one would be able to reproduce the results from their description in the paper.

**Weaknesses:**

The paper's core weaknesses can be narrowed down to 2 key points in the reviewer's eyes: **novelty**, and **strength of results**

Novelty:
* A number of large language model releases have evaluated their models at the smaller scale, and projects like e.g. llamafile go to great lengths to test these smaller models on edge devices, what distinguishes the "Narrow Transformer" from these models aside from being fine-tuned on Java-only?
* The reviewer is left unconvinced that a Java-only fine-tuned transformer is a noteworthy result. While target applications for small language models are alluded to in the _Related Work_ section, they are not reflected in the evaluation design.
* There exist a great number of blogposts out in the web showcasing the fine-tuning of pretrained models on distilled corpora. The reviewer is left unconvinced that this paper in its present state goes beyond this state of the art.

Strength of Results:
* It remains unconvincing to the reviewer from the presented results, that a model fine-tune only on a single programming language (after it was pretrained on a multi-programming language corpus) is superior to a smaller model trained on only Java from the outset. When making such claim, I would expect it to be backed up by experiments by e.g. training a StarCoder-style transformer from scratch on the distilled Java corpus.

**Questions:**

While I understand the premise & aspiration of the authors, I am left to question what the actual research question is in this instance?

Finetuning of pretrained models is almost commoditized by now, and by taking a fine-tuning framework like e.g. Axolotl one could produce similar results fairly quickly if with access to the commensurate compute resources. How do the authors go beyond this, and contribute to small language model development?

What (open-ended) questions I would furthermore like to leave the authors with are:
* Are there specific small model optimizations which could e.g. improve latency, or the ability to deploy the model on tiny devices or even embedded hardware
* Do existing evaluation metrics reflect the specific demands of small language models adequately? If not, which aspects are missing and should see custom evaluation tasks?
* What applications do the authors envision for small language models specifically, and what evaluation design would be derived from these downstream demands?

---

### Meta-Review · Area_Chair_NY8T · 2024-12-22

**Metareview:**

> This work introduced the NT-Java-1.1B model, detailing its development process and evaluation results. NT-Java-1.1B is developed based on the StarCoderBase-1.1B model and trained on a subset of StarCoderData. Evaluation results indicate that NT-Java-1.1B outperforms StarCoderBase-3B in pass@1 performance on the MULTIPL-E benchmark, while it scores lower than StarCoderBase-1.1B on HumanEval-FIM (Java).

The reviewers are unanimous, this paper is not a sufficient contribution for publication at ICLR.

**Additional Comments On Reviewer Discussion:**

There was no rebuttal written by the authors, so no discussion.

---

### Decision · Program_Chairs · 2025-01-22

Reject